

# Arabitol, mannitol and glucose as tracers of primary biogenic organic aerosol: influence of environmental factors on ambient air concentrations and spatial distribution over France

Abdoulaye Samaké[1], Jean-Luc Jaffrezo[1], Olivier Favez[2], Samuël Weber[1], Véronique Jacob[1], Trishalee Canete[1], Alexandre Albinet[2], Aurélie Charron[1,16], Véronique Riffault[3], Esperanza Perdrix[3], Antoine Waked[1], Benjamin Golly[1], Dalia Salameh[1*], Florie Chevrier[1,4], Diogo Miguel Oliveira[2,3], Jean-Luc Besombes[4], Jean M.F. Martins[1], Nicolas Bonnaire[5], Sébastien Conil[6], Géraldine Guillaud[7], Boualem Mesbah[8], Benoit Rocq[9], Pierre-Yves Robic[10], Agnès Hulin[11], Sébastien Le Meur[12], Maxence Descheemaecker[13], Eve Chretien[14], Nicolas Marchand[15], and Gaëlle Uzu[1].

[1]University Grenoble Alpes, CNRS, IRD, INP-G, IGE (UMR 5001), 38000 Grenoble, France
[2]INERIS, Parc Technologique Alata, BP 2, F-60550 Verneuil-en-Halatte, France
[3]IMT Lille Douai, University Lille, SAGE – Département Sciences de l'Atmosphère et Génie de l'Environnement, 59000 Lille, France
[4]University Savoie Mont-Blanc, LCME, 73000 Chambéry, France
[5]LSCE, UMR CNRS-CEA-UVSQ, 91191 Gif-sur Yvette, France
[6]ANDRA DRD/GES Observatoire Pérenne de l'Environnement, F-55290 Bure, France
[7]Atmo Auvergne-Rhône-Alpes, 38400 Grenoble, France
[8]Air PACA, 03040, France
[9]Atmo Hauts de France, 59000, France
[10]Atmo Occitanie, 31330 Toulouse, France
[11]Atmo Nouvelle Aquitaine, 33000, France
[12]Atmo Normandie, 76000, France
[13]Lig'Air, 45590 Saint-Cyr-en-Val, France
[14]Atmo Grand Est, 16034 Strasbourg, France
[15]University Aix Marseille, LCE (UMR7376), Marseille, France
[16]IFSTTAR, F-69675 Bron, France
[*]Now at: Airport pollution control authority (ACNUSA), 75007 Paris, France

*Corresponding author(s):* A Samaké (abdoulaye.samake2@univ-grenoble-alpes.fr) and JL Jaffrezo (Jean-luc.Jaffrezo@univ-grenoble-alpes.fr)



**Abstract**. The primary sugar compounds (SC, defined as glucose, arabitol and mannitol) are widely recognized as
suitable molecular markers to characterize and apportion primary biogenic organic aerosol emission sources. This
work improves our understanding of the spatial behavior and distribution of these chemical species and evidences
their major effective environmental drivers. We conducted a large study focusing on the daily (24 h) $PM_{10}$ SC
concentrations for 16 increasing space scale sites (local to nation-wide), over at least one complete year. These
sites are distributed in several French geographic areas of different environmental conditions. Our analyses, mainly
based on the examination of the short-term evolutions of SC concentrations, clearly show distance-dependent
correlations. SC concentration evolutions are highly synchronous at an urban city-scale and remain well correlated
throughout the same geographic regions, even if the sites are situated in different cities. However, sampling sites
located in two distinct geographic areas are poorly correlated. Such pattern indicates that the processes responsible
for the evolution of the atmospheric SC concentrations present a spatial homogeneity over typical areas of at least
tens of kilometers. Local phenomena, such as resuspension of topsoil and associated microbiota, do no account for
the major emissions processes of SC in urban areas not directly influenced by agricultural activities. The
concentrations of SC and cellulose display remarkably synchronous temporal evolution cycles at an urban site in
Grenoble, indicating a common source ascribed to vegetation. Additionally, higher concentrations of SC at another
site located in a crop field region occur during each harvest periods, pointing out resuspension processes of plant
materials (crop detritus, leaf debris) and associated microbiota for agricultural and nearby urbanized areas. Finally,
ambient air temperature, relative humidity and vegetation density constitute the main effective drivers of SC
atmospheric concentrations.
**1. Introduction**
Primary biogenic organic aerosols (PBOA), which notably comprise bacterial and fungal cells or spores; viruses;
or microbial fragments such as endotoxins and mycotoxins; and pollens and plant debris, are ubiquitous particles
released from the biosphere to the atmosphere (Amato et al., 2017; Després et al., 2012; Elbert et al., 2007; Fang
et al., 2018; Fröhlich-Nowoisky et al., 2016; Morris et al., 2011; Wéry et al., 2017). PBOA can contribute
significantly to the total coarse aerosol mass (Amato et al., 2017; Bozzetti et al., 2016; Coz et al., 2010; Fröhlich-
Nowoisky et al., 2016; Jaenicke, 2005; Manninen et al., 2014; Morris et al., 2011; Samaké et al., 2019; Vlachou
et al., 2018; Yue et al., 2017). Besides their expected negative human health effects (Fröhlich-Nowoisky et al.,
2009, 2016; Humbal et al., 2018; Lecours et al., 2017), they substantially influence the carbon and water cycles at
the global scale, notably acting as cloud and ice nuclei (Ariya et al., 2009; Elbert et al., 2007; Fröhlich-Nowoisky
et al., 2016; Hill et al., 2017; Humbal et al., 2018; Morris et al., 2014; Rajput et al., 2018). While recent studies
have revealed highly relevant information on the abundance and size partitioning of PBOA, their emission sources
and contribution to total airborne particles are still poorly documented, partly due to the analytical limitations to
distinguish PBOA from other types of carbonaceous particulate matter (Bozzetti et al., 2016; China et al., 2018;
Di Filippo et al., 2013; Heald and Spracklen, 2009; Jia et al., 2010). Notably, the global emissions of fungal spore
emitted into the atmosphere are still poorly constrained and range from 8 $Tg.y^{-1}$ to 186 $Tg.y^{-1}$ (Després et al., 2012;
Elbert et al., 2007; Jacobson and Streets, 2009; Sesartic and Dallafior, 2011).
Recently, source-specific tracer methodologies have been introduced to estimate their contribution to aerosol
loadings (Bauer et al., 2008a; Di Filippo et al., 2013; Gosselin et al., 2016; Zhang et al., 2010, 2015). Indeed,
atmospheric organic aerosols (OA) contain specific chemical species that can be used as reliable biomarkers in
tracing the sources and abundance of PBOA (Bauer et al., 2008a; Gosselin et al., 2016; Holden et al., 2011; Jia
and Fraser, 2011; Medeiros et al., 2006b). For instance, sugar alcohols (aka polyols)—including arabitol and
mannitol (two common storage soluble carbohydrates in fungi)—have been recognized as tracers for airborne
fungi, and their concentrations are widely used to estimate PBOA contributions to OA mass (Amato et al., 2017;
Bauer et al., 2008a, 2008b; Golly et al., 2018; Medeiros et al., 2006b; Samaké et al., 2019; Verma et al., 2018;
Weber et al., 2018; Zhang et al., 2010; Zhu et al., 2015, 2016). Similarly, glucose has also been used as a specific



tracer for plant materials (such as pollen, leaves, and their fragments) or soil emissions within various studies
around the world (Chen et al., 2013; Fu et al., 2013; Liang et al., 2016; Medeiros et al., 2006b; Pietrogra,nde et al.,
2014; Rathnayake et al., 2017; Rogge et al., 2007; Simoneit et al., 2004b; Wan and Yu, 2007; Wan et al., 2019).
In this context, atmospheric concentrations of specific polyols and/or primary monosaccharides (including
glucose) have been previously quantified at sites in several continental, agricultural, coastal or polar regions
(Barbaro et al., 2015; Chen et al., 2013; Fu et al., 2012; Golly et al., 2018; Graham et al., 2003; Jia et al., 2010;
Liang et al., 2016; Pietrogra,nde et al., 2014; Rogge et al., 2007; Simoneit et al., 2004a; Verma et al., 2018; Yttri
et al., 2007; Zhu et al., 2018). However, large datasets investigating their (multi)annual cycles, seasonal and
simultaneous short-term variations at multiple spatial scale resolutions (i.e. from local to continental) are still
lacking (Liang et al., 2013; Nirmalkar et al., 2018; Pietrogra,nde et al., 2014; Yan et al., 2019). Such records are
essential to better understand the spatial behavior of primary sugar compound (SC) concentrations (i.e., glucose,
arabitol and mannitol) and PBOA emission processes, and to isolate their potential key drivers (e.g., vegetation
type and density, topography, weather conditions, etc.), which are still unclear (Bozzetti et al., 2016). This
information would be essential for further implementation into chemical transport models (Heald and Spracklen,
2009; Tanarhte et al., 2019).
It is commonly acknowledged that SC (particularly arabitol and mannitol) originate from primary biogenic derived
sources such as bacterial, fungal spores, and plant materials (Di Filippo et al., 2013; Golly et al., 2018; Gosselin
et al., 2016; Graham et al., 2003; Holden et al., 2011; Medeiros et al., 2006b; Simoneit et al., 2004b; Wan et al.,
2019; Yan et al., 2019; Yttri et al., 2007, 2011a; Zhu et al., 2015). Some studies have characterized the composition
of SC in topsoil samples (for fractions larger than $PM_{10}$) from both, natural (i.e., uncultivated) and agricultural
regions (Medeiros et al., 2006a; Rogge et al., 2007; Simoneit et al., 2004b; Wan and Yu, 2007). The authors
suggested that the particulate arabitol, mannitol and glucose are introduced into the atmosphere mainly through
resuspended soils or dust particles and associated biota derived from natural soil erosion, unpaved road dust or
agricultural practices. Conversely, Jia and Fraser (2011) reported higher concentrations of SC relative to PBOA in
size-segregated aerosol samples collected at a suburban site (Higley, USA) compared to the local size-fractionated
soils (equivalent to atmospheric $PM_{2.5}$ and $PM_{10}$). This suggested that direct emissions from biota (microbiota,
vascular plant materials) could also be a significant atmospheric input process for SC at this suburban site.
A large database on SC concentrations was obtained over France in the last decade. It already allowed the
investigation of the size distribution and seasonal variabilities of SC concentrations in aerosols at 28 French sites,
notably showing that SC are ubiquitous primary aerosols, accounting for a significant proportion of $PM_{10}$ organic
matter (OM) mass (Samaké et al., 2019). Results confirmed that their ambient concentrations display a well-
marked seasonality, with maximum concentrations from late spring to early autumn, followed by an abrupt
decrease in late autumn, and a minimum concentration during wintertime in France. This study also showed that
the mean PBOA chemical profile is largely dominated by organic compounds, with only a minor contribution of
dust particle fraction. The latter result indicated that ambient polyols could most likely be associated with direct
biological particle emissions (e.g. active spore discharge, microbiota released from phylloplane or phyllosphere,
etc.) rather than with the microorganism-containing soil resuspension. These observations call for more
investigations of the predominant SC (and PBOA) emission sources.
Cellulose, a linear polymer composed of D-glucopyranose units linked by β-1,4 bonds, is the most frequent
polysaccharide occurring in terrestrial environments (Ramoni and Seiboth, 2016). Plant materials contain cellulose



which has been reported as a suitable proxy to evaluate the vegetative debris contribution to OM mass (Bozzetti
et al., 2016; Glasius et al., 2018; Puxbaum and Tenze-Kunit, 2003; Sánchez-Ochoa et al., 2007; Yttri et al., 2011b).
The ambient $PM_{10}$ cellulose has been shown to be abundant in the European semi-rural or background
environments (accounting for 2 to 10 % of OM mass) (Glasius et al., 2018; Sánchez-Ochoa et al., 2007) and Nordic
rural environments in Norway (contributing to 12 to 18 % of total carbon mass) (Yttri et al., 2011b). Thus,
simultaneous concentration measurements of cellulose and SC can provide essential information into their
emission source dynamics.
As the continuation of our previous work (Samaké et al., 2019), the present paper aims to delineate the processes
that drive the atmospheric concentrations of SC and then PBOA. This is achieved through (i) the analysis of
simultaneous annual short-term time series of particulate SC concentrations over pairs of sites across multiple
space ranges, including local, regional and nationwide sites, and (ii) the investigation of links between
concentrations and series key parameters such as meteorological and phenological ones. Simultaneous annual
short-term concentration measurements of SC and cellulose was performed to better understand of their sources
correlations.
**2. Material and methods**
**2.1 Sampling sites**
Daily $PM_{10}$ concentrations reported in the present work were obtained from different research and monitoring
programs conducted over the last six years in France. Within the framework of the present study, we carefully
selected sites sharing at least one complete year of concurrent monitoring with another one, to be representative
of the annual variation cycles. The final dataset includes data from 16 sites, which are distributed in different
regions of France (Figure 1) and cover several main types of environmental conditions in terms of site topography,
local vegetation, and climate. The characteristics and data available at each sampling site are listed in Table S1 of
the supplementary material (SM), together with the information on the annual average concentrations of aerosol
chemical composition (Table S2). Detailed information on the sampling conditions can be found in Samaké et al.
(2019), such as the campaign periods, number of collected PM samples, sampling flow rates, sample storage and
handling, etc. Note that, the previous database (Samaké et al., 2019) has been updated here with arabitol and
mannitol in $PM_{10}$ collected at the suburban site of Nogent-sur-Oise for a series covering the years 2013 to 2017.

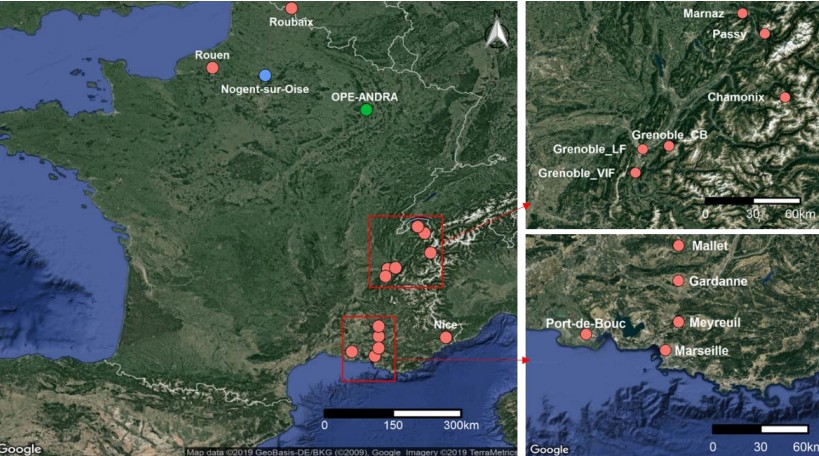



**Figure 1: Geographical location of the selected sampling sites. The red and blue dots indicate respectively urban and**
**suburban sites while the green one corresponds to a rural site, surrounded by field crop areas.**
**2.2 Chemical analyses**
Daily (24 h) $PM_{10}$ samples were collected onto prebaked quartz fiber filter (Tissuquartz PALL QAT-UP 2500 150
mm diameter) every third or sixth day, but not concurrently at all sites. They were then analyzed for various
chemical species using subsampled fractions of the collection filters and a large array of analytical methods. Details
of all the chemical analysis procedures are reported elsewhere (Golly et al., 2018; Samaké et al., 2019; Waked et
al., 2014; Weber et al., 2018). Briefly, primary sugar compounds were extracted from filter aliquots (punches
typically about 10 cm²) into ultrapure water. The extracts are then filtered using a 0.22 µm Acrodisc filter.
Depending on the site, analyses were conducted either by the IGE (Institut des Géosciences de l'Environnement)
or by the LSCE (Laboratoire des Sciences du Climat et de l'Environnement) (Samaké et al., 2019). At the IGE,
extraction was performed during 20 min in a vortex shaker and analyses were achieved using high-performance
liquid chromatography with pulsed amperometric detection (HPLC-PAD). A first set of equipment was used until
March 2016, consisting of a Dionex DX500 equipped with three columns Metrosep (Carb 1-Guard + A Supp 15-
150 + Carb 1-150), the analytical program was isocratic with 70 mM sodium hydroxide (NaOH) as eluent for 11
min, followed by a gradient cleaning step with a 120 mM NaOH as eluent for 9 min. This procedure allows the
analysis of arabitol, mannitol and glucose (Waked et al., 2014). A second set of equipment was used after March
2016, with a Thermo-Fisher ICS 5000+ HPLC equipped with 4 mm diameter Metrosep Carb $2 \times 150$ mm column
and 50 mm pre-column. The analytical run was isocratic with 15 % of an eluent of sodium hydroxide (200 mM)
and sodium acetate (4 mM) and 85 % water, at 1 mL min⁻¹. At the LSCE, extraction was performed for 45 min by
sonication and analyses were achieved using ion chromatography instrument (IC, DX600, Dionex) with Pulsed
Amperometric Detection (ICS3000, Thermo- Fisher). In addition, a CarboPAC MA1 column has been used (4 ×
250 mm, Dionex) along with an isocratic analytical run with 480 mM sodium hydroxide eluent. This analytical
technique allows to quantify arabitol, mannitol and glucose (Srivastava et al., 2018).
For cellulose quantification, we used an optimized protocol based on that described by (Kunit and Puxbaum, 1996;
Puxbaum and Tenze-Kunit, 2003), in which the cellulose contained in the lignocellulosic material is enzymatically
hydrolyzed into glucose units before analysis. Since the alkaline peroxide pretreatment step used to remove lignin
in the original protocol results in a loss of sample material, it has been avoided in this study. Therefore, only the
"free cellulose" is reported in our samples. Note that Sánchez-Ochoa et al., (2007) consider that this free cellulose
could represent only about 70 % of the total cellulose in air samples and that the total cellulose could represent
only about 50 % of the "plant debris" content of atmospheric PM. Very few other results are available on this topic
(Bozzetti et al., 2016; Glasius et al., 2018; Vlachou et al., 2018; Yttri et al., 2011b). The protocol has been improved
to increase sensitivity and accuracy, by reducing the contribution of glucose in the blanks and by using an HPLC-
PAD as the analytical method for the determination of glucose concentrations. Trichoderma reesei cellulase (>700
u g-1, Sigma Aldrich) and Aspergilus Niger glucosidase (>750 u g⁻¹, Sigma Aldrich) have been used as
saccharification enzymes. The protocol is detailed in Section 2 of the SM.
Field blank filters (about 10 % of samples) were handled as real samples for quality assurance. The present data
have been corrected from field blanks. The reproducibility of the analysis of primary sugar compounds (polyols,
glucose) and cellulose, estimated from the analysis of sample extracts from 10 punches of the same filters were in



the range of 10-15 %. About 2 800 samples are considered in this work for the polyols and glucose series, while
290 samples (from the sites of Grenoble_LF and OPE-ANDRA) are considered for the cellulose series.

**2.3 Meteorological data and LAI measurements**

Ambient weather data were not available at all monitoring sites (see Table S1). In this study, data including daily
relative humidity (%), night-time temperature (°C), average and maximum temperatures (°C), wind speed (m s$^{-1}$),
solar radiation (W m$^{-2}$), and rainfall level (mm) for the sites of Marnaz and OPE-ANDRA (Figure 1), representing
different climatic regions and environmental conditions, were obtained from the French meteorological data
sharing service system (Météo-France) and ANDRA (French national radioprotective agency, in charge of the
OPE-ANDRA site), respectively.
The leaf area index (LAI), which is defined as the projected area of leaves over a unit of land, is an important
measure of the local vegetation density variation (Heald and Spracklen, 2009; Yan et al., 2016a, 2016b). For this
study, we used the MODIS Collection 6 LAI product because it is considered to have the highest quality among
all the MODIS LAI products (Yan et al., 2016a, 2016b). The MCD15A3H product uses both Terra and Aqua
reflectance observations as inputs to estimate daily LAI at 500 m spatial resolution, and a 4-day composite is
calculated to reduce the noise from abiotic factors. Using a $2 \times 2$ km grid box around the monitoring site, the local
vegetation density variation was retrieved from LP DAAC (https://lpdaac.usgs.gov/, last accessed: 15 March 2019)
for the sites of Marnaz, OPE-ANDRA, and Grenoble_LF.

**2.4 Data analyses**

All the statistical analyses were carried out using the open-source R software (R studio interface, version 3.4.1).
Several statistical analyses were performed on the concentrations to identify the spatial patterns of emission
sources and the potential parameters of influence as explained below.
The normalized cross-correlation (NCC) test was chosen to examine the potential similarities among the
monitoring sites for particulate SC concentrations, in terms of short-term temporal trends (e.g. synchronized
periods of increase or decrease, simultaneous fluctuations during specific episodes). The main advantage of NCC
over the traditional correlation tests is that it is less sensitive to linear changes in the amplitudes of the two-time
series compared. Therefore, to reduce the possibility of spurious "anti-correlation" due to highly variable
concentration ranges, data were amplitude-normalized prior to correlation analysis. A thorough discussion on the
normalized cross-correlation method can be found elsewhere (Kaso, 2018; Yoo and Han, 2009). To achieve pair-
wise correlation analysis between the sampling sites collected during the same periods, the original daily
measurements were processed as follows: starting on identical days, arrangement on the original daily data into
consecutive 3-day intervals (or 6-day intervals in the case of OPE-ANDRA) and calculation of the average
concentration values for the middle-day were performed. The resultant data were used for correlation analysis
(Table S3).
Multiple linear regression (MLR) was used to assess the strength of the relationships between atmospheric
concentrations of particulate SC and local environmental factors including the daily mean relative humidity, night-
time temperature, average and maximum temperature, wind speed, solar radiation, rain levels and LAI. Because
the LAI is a 4-day composite, daily values of the other variables were re-scaled into consecutive 4-day averaged
values. The linear regression (lm) package in R was employed for multiple regression analyses. The concentration
data were log-transformed to obtain regression residual distributions as close as possible to the normal Gaussian





one (Figure S1). Stepwise forward selection was used to select the predictors that explain well the temporal
variation of SC concentrations at the site of Marnaz.
It should be noted that due to the limited availability of external parameters, the environmental factors driving SC
atmospheric levels have been extensively investigated for only two monitoring sites with contrasted
characteristics: the urban background site of Marnaz located in an Alpine valley, and the rural OPE-ANDRA site
surrounded by field crop areas spreading over several tens of km.

**3. Results and discussion**
**3.1 Example of spatial coherence of the concentrations at different scales**
Our previous work (Samaké et al., 2019) showed that particulate polyols and glucose are ubiquitous primary
compounds with non-random spatial and seasonal variation patterns over France. Here, an inter-site comparison
of their short-term concentration evolutions has been carried out at different space scales (from local to national)
for the pairs that can be investigated in our data base. Figure 2 presents some of these comparisons for 3 spatial
scales (15, 120, and 205 km).
The daily average concentrations of polyols (defined as sum of arabitol and mannitol) and glucose display highly
synchronous evolutional trends (i.e., homogeneity in the concentrations, the timing of concentration peaks,
simultaneity of the daily specific episodes of increase/decrease of concentrations) over 3 neighboring monitoring
sites located 15 km apart in the Grenoble area (Figures 2A and B). Interestingly, remarkable synchronous patterns
both for short term (near-daily) and longer term (seasonal) still occur for sites located 120 km apart, as exemplified
for 2 sites in Alpine environments (Grenoble and Marnaz) (Figures 2C and D). However, as shown in Figures 2E
and F, the evolutions of concentrations become quite dissimilar and asynchronous in terms of seasonal and daily
fluctuations for more distant sites (Grenoble and Nice, 205 km apart), that are located in different climatic regions
(Alpine for Grenoble, Mediterranean for Nice). This is contrasting with results from the rural background site of
OPE-ANDRA and the suburban site of Nogent-sur-Oise, both located in a large field crop region of extensive
agriculture, and about 230 km apart from each other (Figure 2G). Indeed, they present very similar variations of
daily concentrations for multi-year series, despite their distance apart, with concentration peaks generally more
pronounced at the rural site of OPE-ANDRA.
The following sections are dedicated to the investigation of the processes that can lead to these similarities and
differences according to these spatial scales.





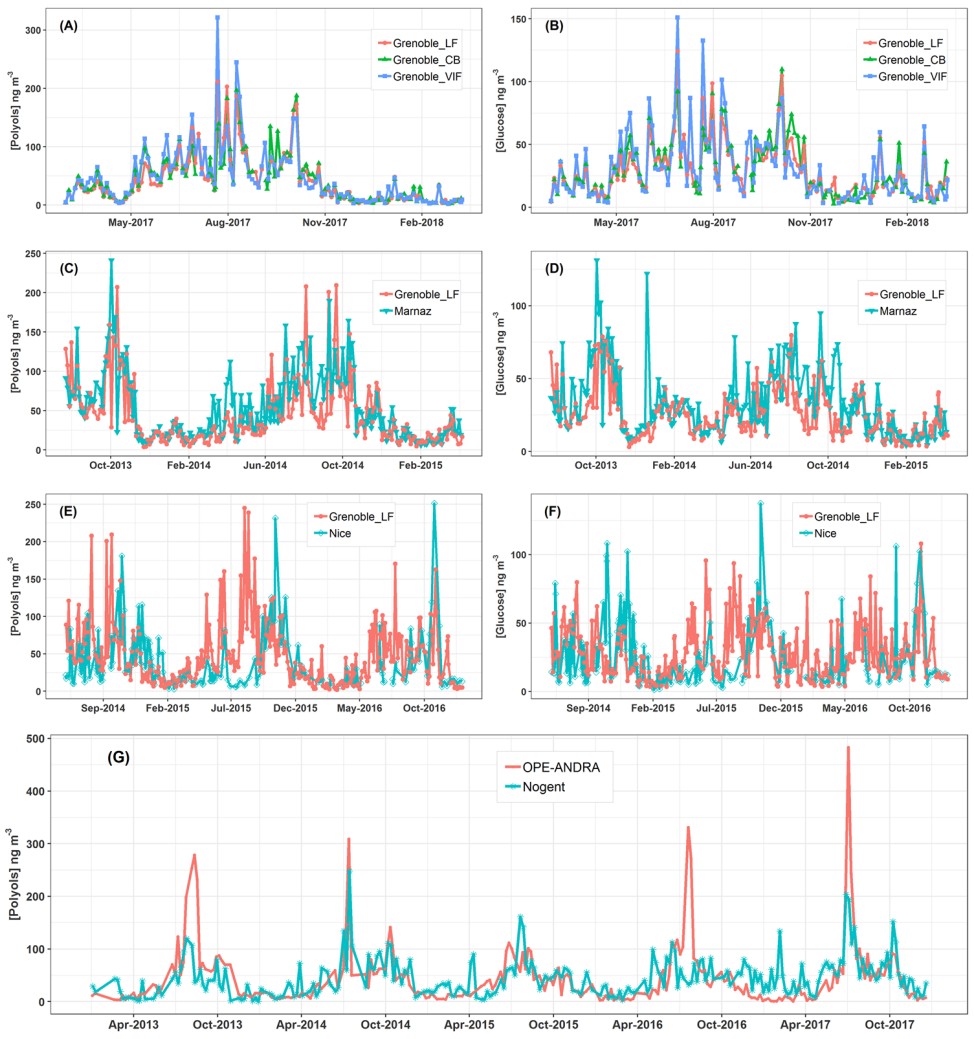


**Figure 2: Concentrations (in ng m⁻³) of (left) ambient particulate polyols (defined as the sum of arabitol and mannitol)**
**and glucose (right) over different monitoring sites in France. Since PM₁₀ were collected every 3-days at Nogent-sur-Oise**
**and 6-days at OPE-ANDRA, the original data sets are averaged over consecutive 6-day intervals (bottom graph).**


**3.2 Inter-site correlations and spatial scale variability**
Figures 3A and 3B provide an overview of the cross-correlation coefficients for the daily evolution of
concentrations (for glucose and polyols (SC)) between pairs of sites located at multiple increasing space scales
across France (Table S3). Time series of concentrations for both SC show a clear distance-dependent correlation.
The strength of the correlations is highly significant for distances up to 150-190 km (R > 0.72, p < 0.01) and
gradually decreases with increasing inter-site distances. One exception is the pair OPE-ANDRA and Nogent-sur-
Oise (high correlation for a distance above 230 km), both sites being located in highly-impacted agricultural areas.
This overall pattern suggests that the processes responsible for the atmospheric concentrations of SC present a
spatial homogeneity over typical areas of at least several tens of km



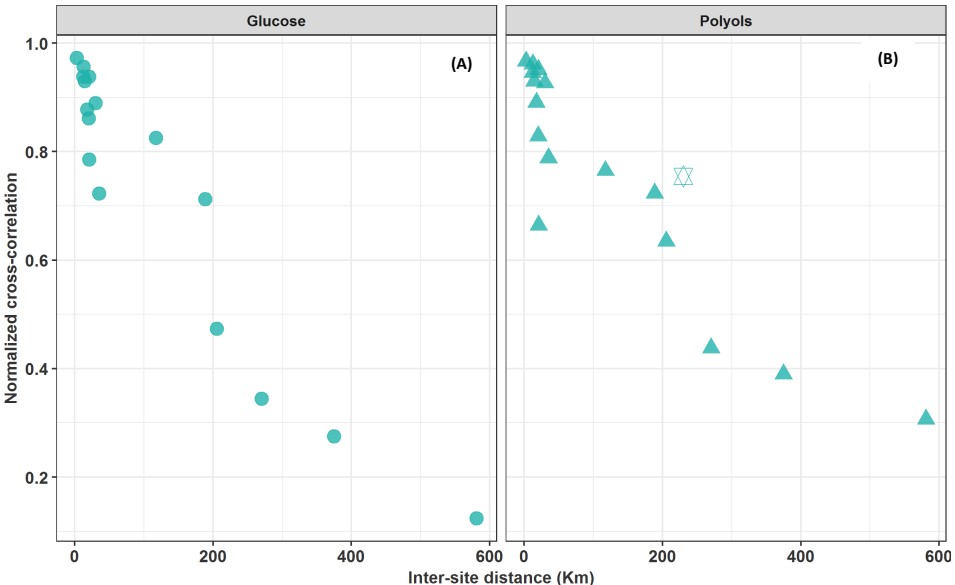


**Figure 3: Normalized cross-correlation values for the daily evolution of particulate glucose (A) and polyols (B) concentrations over pairs of sites located at multiple increasing space scales across France. The hexagram corresponds to the correlation between the sites of OPE-ANDRA and Nogent-sur-Oise, both sites being surrounded by crop field areas.**

Unlike SC, ambient air concentrations of sulfate, associated with long-range aerosol transport (Abdalmogith and
Harrison, 2005; Amato et al., 2016; Coulibaly et al., 2015; Pindado and Perez, 2011; Waked et al., 2014) display
stronger positive correlations (R > 0.72-0.98, p < 0.01) at all pairs of sites considered in the present work (Figure
S2). Moreover, ambient concentrations of calcium, associated with local fugitive dust sources or/and long-range
aerosol transport (Ram et al., 2010; Wan et al., 2019) display random correlation patterns (Figure S2). These results
are in agreement with Zhu et al. (2018) who also reported non-significant correlations between SC and sulfate in
$PM_{2.5}$ aerosols measured at Shanghai, China. The distinct spatial behaviors between sulfate (or $Ca^{2+}$) and SC in the
present work further suggest a dominant regional influence for atmospheric SC, as opposed to processes associated
with either local sources for calcium or long-range transport for sulfate.
Mannitol and arabitol are well-known materials of fungal spores, serving as osmo-regulatory solutes (Medeiros et
al., 2006b; Simoneit et al., 2004b; Verma et al., 2018; Zhang et al., 2010, 2015). Based on parallel measurements
of spore counts and $PM_{10}$ polyol concentrations at three sites within the area of Vienna (Austria), Bauer et al.
(2008a) found an average arabitol and mannitol content per fungal spores of respectively 1.2 pg spore$^{-1}$ (range 0.8-
1.8 pg spore$^{-1}$) and 1.7 pg spore$^{-1}$ (range 1.2-2.4 pg spore$^{-1}$). Mannitol and arabitol have also been often identified
in the green algae and lower plants (Buiarelli et al., 2013; Di Filippo et al., 2013; Vélëz et al., 2007; Xu et al.,
2018; Zhang et al., 2010). Being important chemical species for the metabolism of these microorganisms
(Shcherbakova, 2007), it may well be that the concentration ratio of mannitol-to-arabitol could deliver some
information on the spatial or temporal evolution of their emission processes (Gosselin et al., 2016). The annual
average mannitol-to-arabitol ratio at all sites is about 1.15 ± 0.59, with ratios for the warm period (Jun-Sept) being
1 to 2 times higher than those in the cold period (Dec-May) (Table S1). These ratios are within the range of those
previously reported for $PM_{10}$ aerosols collected at various urban and rural background sites in Europe (Bauer et



al., 2008a; Yttri et al., 2011b). Similarly, Burshtein et al., (2011) also reported comparable ratios for $PM_{10}$ aerosols
collected during autumn and winter from a Mediterranean region in Israel.
Similarly, the annual average glucose-to-polyols ratio at all sites is about $0.79 \pm 0.77$. No literature data are
currently available for comparison. Further work is needed to relate these variations with microorganism
communities and plant growing stages.
However, as evidenced in Figure 4, both mannitol-to-arabitol and glucose-to-polyols ratios show a clear distance-
dependent correlation, with higher correlations ($R = 0.64$ to $0.98$, $p < 0.01$) observed for pairs of sites within 150-
190 km distance. This spatial consistency highlights once again that the dominant emission processes should be
effective regionally, rather than being specific local input processes, and that atmospheric dynamics of the
concentration levels (i.e., driven by the interplay of emission and removal processes) are determined by quite
similar environmental factors (e.g. meteorological conditions, vegetation, land use, etc.) at such a regional scale.
This implies that local events and phenomena, such as the mechanical resuspension of topsoil and associated biota
(like bacteria, fungi, plant materials, etc.) might not be their major atmospheric input processes, particularly in
urban background areas typically characterized by less bare soil, and with a variable nature of the unpaved topsoil
at the regional scale (Karimi et al., 2018). Furthermore, Karimi et al. (2018) also recently reported heterogeneous
topsoil microbial structure within patches of 43 to 260 km across different regions of France. It follows that the
hypotheses of emissions related to mechanical resuspension of topsoil particles and associated biota, or microbiota
emitted actively from surface soil into the air generally assumed in most pioneering reports (Medeiros et al., 2006b;
Rogge et al., 2007; Simoneit et al., 2004b; Wan and Yu, 2007) are most probably not valid.
Alternatively, the vegetation leaves have also been suggested as sources of atmospheric SC (Golly et al., 2018; Jia
and Fraser, 2011; Pashynska et al., 2002; Sullivan et al., 2011; Verma et al., 2018; Wan et al., 2019). In fact,
vascular plant leaf surfaces is an important habitat for endophytic and epiphytic microbial communities (Kembel
and Mueller, 2014; Lindow and Brandl, 2003; Whipps et al., 2008). Our results are more in agreement with a
dominant atmosphere entrance process closely linked to vegetation, which is more homogeneous than topsoil at
the climatic regional scale. Consistent with this, Sullivan et al. (2011) also observed evident distinct regional
patterns for daily $PM_{2.5}$ polyols and glucose concentrations at ten urban and rural sites located in the upper Midwest
(USA). The authors attributed such a spatial pattern to the differences in vegetation types and microbial diversity
over distinct geographical regions. Accordingly, the vegetation structure and composition have been previously
shown to play essential roles on airborne microbial variabilities in nearby areas (Bowers et al., 2011;
Lymperopoulou et al., 2016; Mhuireach et al., 2016).

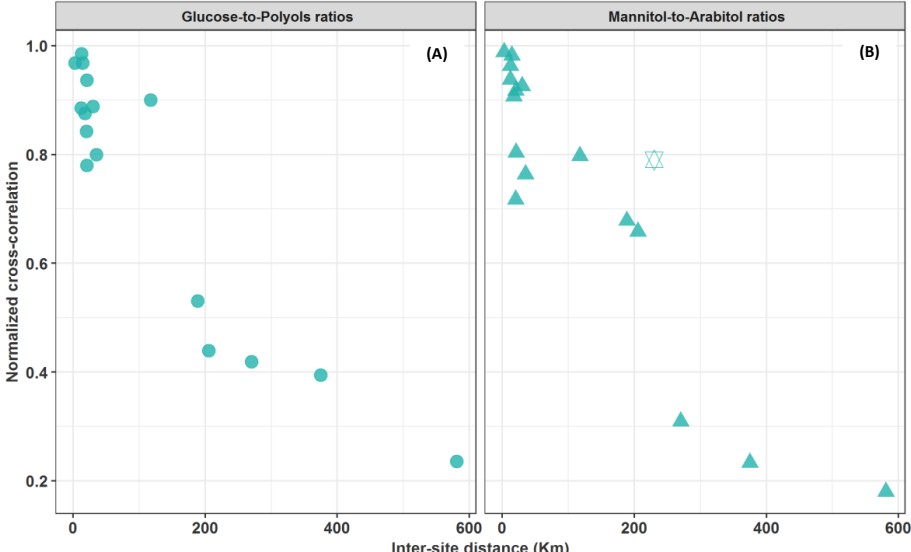


Figure 4: Normalized cross-correlation values for daily evolution of particulate glucose-to-polyols (A) and mannitol-to-arabitol (B) ratios over pairs of sites located at multiple increasing space scales across France. The hexagram corresponds to the correlation between the sites of OPE-ANDRA and Nogent-sur-Oise, both sites being surrounded by crop field areas.

### 3.3 Influence of the vegetation on polyols and glucose concentrations

The relationships between SC $PM_{10}$ concentrations and vegetation (plant materials) can be examined at the site of
Grenoble Les Frênes (Grenoble_LF) by comparing the annual evolutions of SC and the free atmospheric cellulose
concentrations, together with LAI ones.
The daily ambient concentration levels of SC and cellulose range respectively from 5.0 to 301.9 ng m$^{-3}$ (with an
average of $41.2 \pm 39.9$ ng m$^{-3}$) and 0.7 to 207.2 ng m$^{-3}$ (with an average of $52.9 \pm 44.2$ ng m$^{-3}$), which corresponds
to respectively to 0.1 to 6.6 % and 0.01 to 5.3 % of total organic matter (OM) mass in $PM_{10}$. These values are
comparable to those previously reported for various sites in Europe (Daellenbach et al., 2017; Sánchez-Ochoa et
al., 2007; Vlachou et al., 2018; Yttri et al., 2011b). Thus, a major part of PBOA could possibly be ascribed cellulose
and SC derived sources.
As evidenced in Figure 5A, ambient free cellulose concentrations vary seasonally, with maximum seasonal average
values observed in summer ($81.4 \pm 47.6$ ng m$^{-3}$) and autumn ($64.2 \pm 49.2$ ng m$^{-3}$), followed by spring
($52.6 \pm 37.8$ ng m$^{-3}$), and lower levels in winter ($23.0 \pm 19.9$ ng m$^{-3}$). This is the same global pattern for polyols,
that are also more abundant in summer ($82.4 \pm 47.4$ ng m$^{-3}$) and autumn ($48.7 \pm 41.6$ ng m$^{-3}$), followed by spring
($24.9 \pm 16.3$ ng m$^{-3}$), and winter ($10.2 \pm 9.6$ ng m$^{-3}$) in the Grenoble area. On a daily scale, the episodic increases
or decreases of polyols in $PM_{10}$ are very often well synchronized with that of cellulose (figure 5A). Moreover, the
maximum atmospheric concentrations of polyols also mainly occur when the vegetation density (LAI) is at its
highest in late summer (Figure 5B). Similar global behaviors are also observed for atmospheric particulate glucose
and LAI (Figs. 5A and B). To further assess the relationships between SC $PM_{10}$ concentrations and vegetation at
a rural area, a two-year measurement of cellulose concentrations at the highly-impacted agricultural rural site of
OPE-ANDRA has been conducted. The average concentration of cellulose at OPE-ANDRA ($197.9 \pm 217.8$ ng m$^{-}$





³) is 3.5 times higher than that measured in the urban area of Grenoble. In terms of temporal dynamics, the
evolution cycles (i.e., peaks and decreases) of both polyols and glucose are also very often well synchronized with
that of cellulose at OPE-ANDRA (Fig. 5C).
Altogether, these findings highlight that particulate SC $PM_{10}$ and cellulose in both urban background and rural
agricultural areas most probably share a common source related to the vegetation. This is an additional evidence
in support of the hypothesis suggested in previous studies (Bozzetti et al., 2016; Burshtein et al., 2011; Daellenbach
et al., 2017; Pashynska et al., 2002; Verma et al., 2018; Vlachou et al., 2018; Yttri et al., 2007). It is also in line
with studies indicating that the PBOA source profile identified using offline aerosol mass spectrometry (offline-
AMS) correlates very well with coarse cellulose concentrations (Bozzetti et al., 2016; Vlachou et al., 2018).
Noticeable contribution of cellulose to PBOA mass (26 %) at the rural background site of Payerne (Switzerland),
during summer 2012 and winter 2013, was reported by (Bozzetti et al., 2016).
As also evidenced in Figure 5, the cellulose concentration peaks are not systematically correlated to those of
polyols. The development stage of the plants (developing or mature leaves, flowering plants) in addition to the
metabolic activities of endophytic and epiphytic biota (growth, sporulation), all closely related to meteorological
conditions (Bodenhausen et al., 2014; Bringel and Couée, 2015; Lindow and Brandl, 2003; Moricca and Ragazzi,
2011; Reddy et al., 2017), could explain such observations. The influence of local meteorological conditions for
an urban Alp valley site is discussed in Section 3.4. Consistent with our observations, previous studies conducted
at various urban background sites in Europe have suggested that particulate polyols are associated to mature plant
leaves and microorganisms (bacterial and fungal spores) while glucose, which is a monomer of cellulose, would
most likely be linked to the developing leaves (Bozzetti et al., 2016; Burshtein et al., 2011; Pashynska et al., 2002;
Yttri et al., 2007).



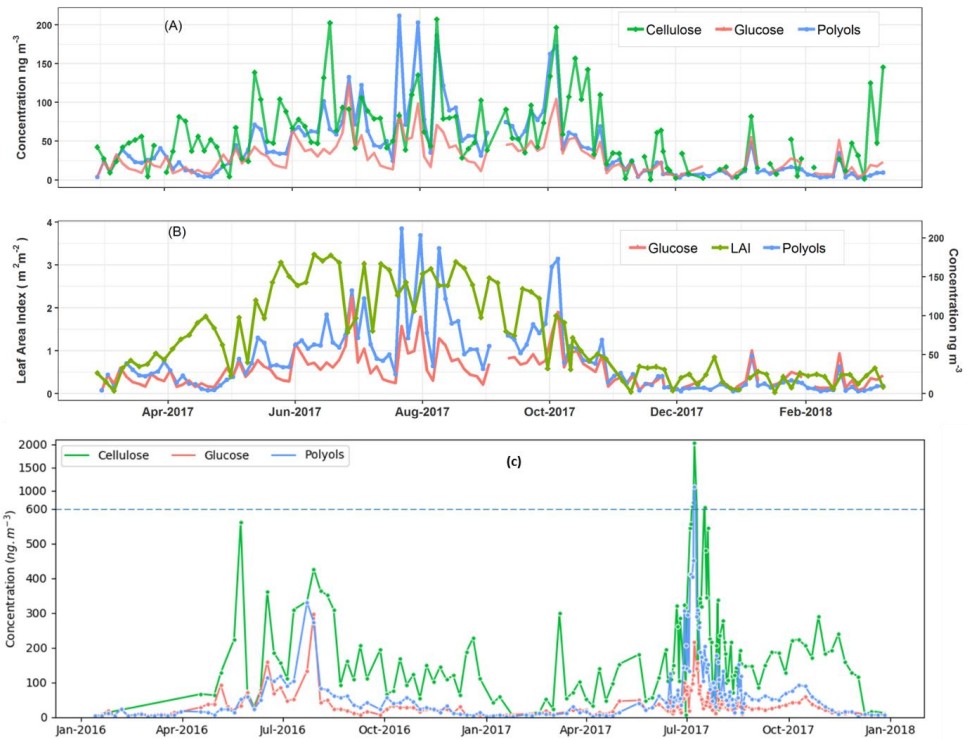

**Figure 5: Temporal covariation cycles of the daily particulate polyols and glucose concentrations along with vegetation indicators at the urban background site of Grenoble (A and B) and the rural agricultural background site of OPE-ANDRA (C), respectively. Note that PM$_{10}$ aerosols are intensively collected at OPE-ANDRA every day (24-h) from 12 June 2017 to 22 August 2017, and that the concentration scale is changing above 600 ng m$^{-3}$ in Figure C, due to extreme concentration peak in July 2017.**

**3.4 Influence of meteorological parameters on ambient concentrations of polyols and glucose**

We used here a multiple linear regression analysis (MLR) approach to gain further insight about the environmental factors influencing the annual and short time variation cycles of atmospheric SC concentrations. This tentative MLR analysis is focused on the urban background site of Marnaz only since meteorological and other data are readily available for this site and are not influenced too much by some large city effects. Several variables were tested, that are already mentioned in the literature as drivers of SC concentrations. It includes the ambient relative humidity, rainfall level, wind speed, solar radiation, night-time temperature, average (or maximum) temperature, and LAI. Night-time temperature was selected since the time series in Marnaz and Grenoble indicate that the major drop of concentrations in late fall (Figure 2C) is related to the first night of the season with night-time temperature below 5°C. The use of the night-temperature is also consistent with the bi-modal distribution of polyols during night and day time found in previous studies (Claeys et al., 2004; Graham et al., 2003).

Overall, the environmental factors including the mean night-time temperature, relative humidity, wind speed and the leaf area index explain up to 82 % (adjusted R$^2$ = 0.82, see Table 1) of the annual temporal variation cycles of SC concentrations. The mean night-time temperature and LAI contribute respectively to 54 % and 37 % of the observed annual variabilities of SC concentrations. The atmospheric humidity is also a driver for these chemical species (3 % of the explained variation). These results are consistent with previous studies showing that





concentrations of mannitol (in both $PM_{10}$ and $PM_{2.5}$ size fractions) linearly correlate best with the LAI, atmospheric
water vapor and temperature (Heald and Spracklen, 2009; Hummel et al., 2015). All of these drivers have been
previously shown to induce the initial release and influence the long-term airborne microbial (i.e. bacteria, fungi)
concentrations (China et al., 2016; Elbert et al., 2007; Grinn-Gofroń et al., 2019; Jones and Harrison, 2004;
Rathnayake et al., 2017; Zhang et al., 2015).
Besides, the wind speed (range of 0.2 to 5.6 m s⁻¹) seems an additional effective driver affecting the contribution
of the local vegetation to SC concentrations in the atmosphere. Albeit enough air movement is required to passively
release microorganisms along with plant debris into the atmosphere, strong air motions induce higher dispersion.
These observations are in good agreement with those previously reported (Jones and Harrison, 2004; Liang et al.,
2013; Zhang et al., 2010, 2015; Zhu et al., 2018). For instance Liang et al. (2013) have found a negative correlation
between wind speed and polyols concentrations, and the highest atmospheric fungal spores concentrations were
observed for a wind speed range of 0.6 to 1.0 m s⁻¹.
**Table 1: Multiple linear regression for ambient polyols and glucose concentrations and their effective environmental**
**factors at the Marnaz site. Contributions of predictor are normalized to sum 1. "Relaimpo package under R" was**
**used to compute bootstrap confidence intervals for importance of effective predictors (n=1000) (Grömping, 2006).**

|  | *Dependent variable* | *Variability explained by effective predictors* |
|---|---|---|
|  | log(Polyols + Glucose) |  |
| Night-time temperature (°C) | 0.112*** (0.090, 0.133) | 0.538 (0.453, 0.604) |
| Relative Humidity (%) | 0.017*** (0.005, 0.030) | 0.030 (0.018, 0.067) |
| Leaf Area Index | 0.386** (0.034, 0.737) | 0.372 (0.286, 0.444) |
| Wind speed (m s⁻¹) | 0.226 (-0.203, 0.655) | 0.021 (0.015, 0.058) |
| Leaf Area Index × Wind Speed[a] | -0.596*** (-1.001, -0.191) | 0.039 (0.014, 0.085) |
| Constant | 2.023*** (0.787, 3.260) |  |
| Observations | 87 |  |
| $R^2$ | 0.837 |  |
| Adjusted $R^2$ | 0.824 |  |
| Residual Std. Error | 0.297 (df = 81) |  |
| F Statistic | 66.677*** (df = 5; 81) |  |
| Note | **p < 0.01; ***p < 0.001 | [a] stands for interaction between predictors |

375 .

One of the limitations of this study is that 4-day averaged observations do not allow to evaluate the driver
contributions that might explain some short term events for which the influence of meteorological parameters such
as rainfall or solar radiation could also be significant (Grinn-Gofroń et al., 2019; Heald and Spracklen, 2009; Jones
and Harrison, 2004). However, such simple parameterizations could be a first step in considering SC
concentrations in CTM models, and further work is required in this direction in order to generate a robust
parametrization of the emissions.
**3.5 Specific case of a highly-impacted agricultural area**
This section focuses on evidencing the environmental drivers of $PM_{10}$ SC concentrations specific to agricultural
areas. To achieve this objective, the site of OPE-ANDRA has been selected because it is extensively impacted by
agricultural activities, without being too prone to influences by other sources. OPE-ANDRA is a specific rural
background observatory located at about 230 km east of Paris at an altitude of 293 m. It is characterized by a low
population density (< 22 inhabitants km⁻² within an area of 900 km²), with no surrounding major transport road or
industrial activities. The air monitoring site itself lies in a "reference sector" of 240 km², in the middle of a field
crop area (tens of kilometers in all directions). The daily agricultural practices within this reference sector are



recorded and made available by ANDRA. The parcels within the agricultural area are submitted to a 3-year crop-
rotation system. The major crops are wheat, barley, rape, pea and sunflower. Additionally, OPE-ANDRA is also
characterized by a homogeneous type of soil, with a predominance of superficial clay-limestone.
Figure 6 shows the daily evolution of polyols concentrations in the $PM_{10}$ fraction at OPE-ANDRA from 2012 to
2018, together with the agricultural activities recorded daily and averaged over 12 days.
Although the concentration of polyols fluctuates from a year to another, they display clear annual variation cycles,
with higher values in the warm periods (Jun. to Nov.) and lower concentration values in the cold periods (Oct. to
May). Interestingly, the annual concentrations of polyols in 2015 (4.2-111.7 ng m$^{-3}$; annual average:
$37.0 \pm 29.1$ ng m$^{-3}$) are significantly lower than those observed for the other years (0.6-1084.6 ng m$^{-3}$; annual
average: $62.9 \pm 96.8$ ng m$^{-3}$). Similar inter-annual evolution trends, but with variable intensities, are also observed
for glucose concentrations (Figure 6). Year 2015 has been found to be particularly hot and dry at OPE-ANDRA
(Figure 7) whereas the local averaged wind conditions are quite stable over the years within the period of study,
suggesting that the wind conditions are not the main driver of the observed inter-annual variability. These results
highlight that ambient air temperature and humidity are key meteorological drivers of the annual variation cycles
of polyols and glucose concentrations. Hot and dry ambient air conditions may decrease the metabolic activity of
the microorganisms (e.g. microbial growth and sporulation) (Fang et al., 2018; Liang et al., 2013; Meisner et al.,

406    2018).

Finally, maximum ambient concentration levels for both SC and cellulose are observed in excellent temporal
agreement with the harvest periods (late summer) at the ANDRA-OPE site (Figure 6). Harvesting activities have
been previously reported as the major sources for particulate polyols and glucose to the atmosphere in agricultural
and nearby urbanized areas (Golly et al., 2018; Rogge et al., 2007; Simoneit et al., 2004b). Hence, the resuspension
of plant materials (crop detritus, leaves debris) and associated microbiota (e.g., bacteria, fungi) originating from
cultivated lands are most-likely major input processes of $PM_{10}$ polyols and glucose at field crop sites.

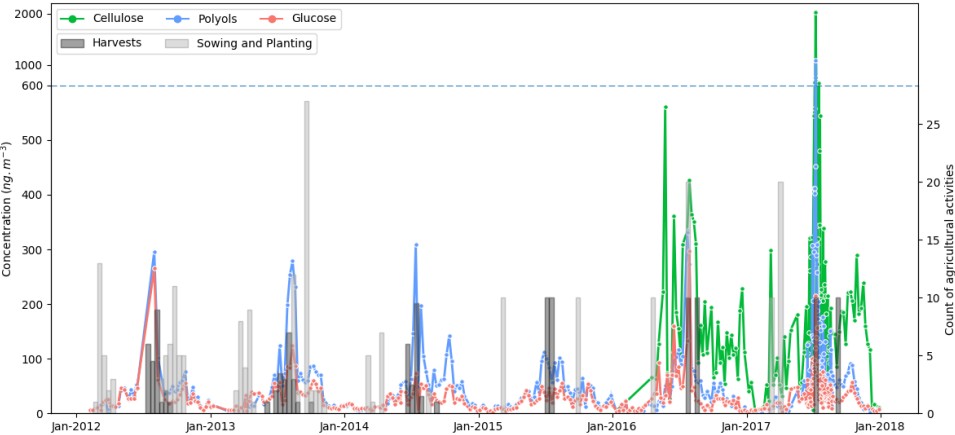


**Figure 6: Daily evolution cycles of polyols and glucose concentrations in aerosols collected from the OPE-ANDRA**
**monitoring site, from 2012 to 2018. Cellulose concentrations have been measured from January 2016 to January 2018.**
**Colored bars correspond to the sum of the various agricultural practices performed (data for 69 parcels are averaged**
**over 12 days for better clarity). Records of agricultural activities after October 2014 were available for only two parcels**
**within the immediate vicinity of the $PM_{10}$ sampler. Records are multiplied by 10 for this period.**





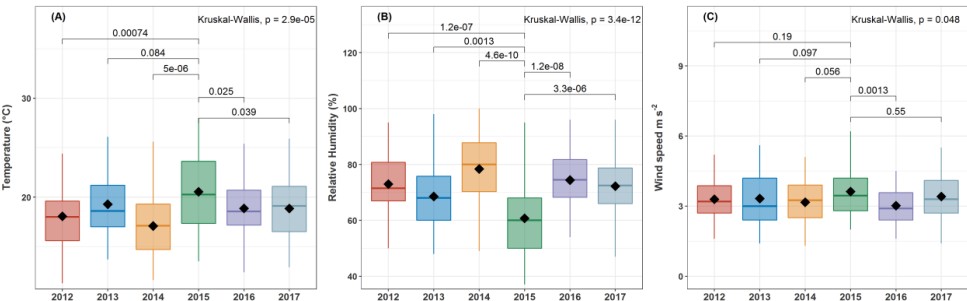

**419**

**Figure 7: Boxplots of (A) maximum ambient temperature, (B) relative humidity and (C) wind speed at OPE-ANDRA from 2012 to 2017. Analyses are performed for warmer periods (June to November). Only statistically different meteorological factors are presented. The black marker inside each boxplot indicates the average value, while the top, middle and bottom of the box represent the 75th, median and 25th percentiles, respectively. The whiskers at the top and bottom of the box extend from the 95th to the 5th percentiles. Statistical differences between average values were assessed with the Kruskall-Wallis method ($p < 0.05$).**

## 4. Conclusions

The short-term temporal (daily) and spatial (local to nation-wide) evolutions of particulate polyols and glucose concentrations are rarely discussed in the current literature. The present work aimed at investigating the spatial behavior of these chemicals and evidencing their major effective environmental drivers. The major results mainly showed that:

- The short-term evolution of ambient polyols and glucose concentrations is highly synchronous across an urban city-scale and remains very well correlated throughout the same geographic areas of France, even if the monitoring sites are situated in different cities at about 150-190 km. However, sampling sites located in two distinct geographic areas are poorly correlated. This indicates that emission sources of these chemicals are uniformly distributed, and their accumulation and removal processes are driven by quite similar environmental parameters at the regional scale. Therefore, local phenomena such as atmospheric resuspension of topsoil particles and associated microbiota, microbial direct emissions (e.g. sporulation), cannot be the main emission processes of particulate polyols and glucose in urban areas not directly influenced by agricultural activities.

- The atmospheric concentrations of polyols (or glucose) and cellulose display remarkably synchronous temporal evolution cycles at the background urban site of Grenoble, indicating a common source related to plant debris.

- Higher ambient concentrations of polyols and glucose at the rural site of OPE-ANDRA occur during each harvest period, pointing out resuspension processes of plant materials (crop detritus, leaves debris) and associated microbiota for agricultural and nearby urbanized areas. This is associated with higher $PM_{10}$ cellulose concentration levels, as high as 0.4 to 2.0 µg.m$^{-3}$ on a daily basis (accounting up to 7.5 to 32.4 % of the OM mass).

- Multiple linear regression analysis of the yearly series from the site of Marnaz gave insightful information on which parameter controls the ambient concentrations of polyols and glucose. Ambient air night-time temperature, relative humidity and vegetation density are the most important drivers, whilst wind speed conditions tend to affect the contribution of local vegetation.





Altogether, these results improve our understanding of the spatial behavior tracers of $PM_{10}$ PBOA emission sources
in France, and in general, which is imperative for further implementation of this important mass fraction of OM
into chemical transport models. Further investigations of airborne microbial fingerprint (bacteria and fungi) are
ongoing, which may deepen our understanding of the PBOA source profile.
**Acknowledgements:** We would like to express special acknowledgements to Pierre Taberlet (LECA, Grenoble,
France) for fruitful discussions about the importance of endophytic and epiphytic biota for aerobiology. The PhD
of AS and SW are funded by the Government of Mali and ENS Paris, respectively. We gratefully acknowledge
the LEFE-CHAT and EC2CO programs of the CNRS for financial supports of the CAREMBIOS multidisciplinary
project, and the LEFE-CHAT program for the MECEA project for the development of the atmospheric cellulose
measurements. Samples were collected and analyzed in the frame of many different programs funded by ADEME,
Primequal, the French Ministry of Environment, the CARA program led by the French Reference Laboratory for
Air Quality Monitoring (LCSQA), ANDRA, and actions funded by many AASQA, IMT Lille Douai (especially
Labex CaPPA ANR-11-LABX-0005-01 and CPER CLIMIBIO projects). Analytical aspects were supported at
IGE by the Air-O-Sol platform within Labex OSUG@2020 (ANR10 LABX56). We acknowledge the work of
many engineers in the lab at IGE for the analyses (Aude Wack, Céline Charlet, Fany Donaz, Fany Masson, Sylvie
Ngo, Vincent Lucaire, Claire Vérin, and Anthony Vella). Finally, the authors would like to kindly thank the
dedicated efforts of many other people at the sampling sites and in the laboratories for collecting and analyzing
the samples.
**Author contributions:** JLJ was the (co-)supervisor for the PhD for AS, FC, SW, and for the post-doc of DS,
BG, and AW. He directed all the personnel who performed the analysis at IGE. He is the coordinator for the CNRS
LEFE-EC2CO CAREMBIOS program that is funding the work of AS. GU and JMF-M were the co-supervisor for
the PhD of AS or SW. EP, OF, and VR supervised the PhD of DMO who investigated the sites in northern France.
OF, JL-J, JL-B, AA and NM were coordinating and partners of the different initial programs for the collection and
chemical analysis of the samples. VJ developed the analytical techniques for polyols and cellulose measurements.
TC performed the cellulose measurements. Samples analyses at LSCE were performed by NB. AC gave advices
for the statistical aspects of the data processing. AS and JLJ processed the data and wrote up the manuscript. SW
participated to the visualization of the results. SC is supervising the OPE station and collected the agricultural
activities records. All authors from AASQA (author affiliation nos. 7 to 14) are representatives for each network
that conducted the sample collection and the general supervision of the sampling sites. All authors reviewed and
commented on the manuscript.
**Competing interests:** The authors declare that they have no conflict of interest.

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
