# Peer review of "Arabitol, mannitol and glucose as tracers of primary biogenic organic aerosol: influence of environmental factors on ambient air concentrations and spatial distribution over France"

_Atmospheric Chemistry and Physics, 2019_

## Referee Comment (RC1) · Anonymous Referee #1 · 18 Jun 2019

This manuscript provides detailed insights into the biogenic primary organic aerosol emission sources of the primary sugar compounds (SC), i.e., glucose, arabitol and mannitol. The study has been carefully designed and the results have been interpreted in detail. The study covers 16 nation-wide sites all over France and contains a very comprehensive data set. It is clearly shown that the main drivers of SC atmospheric concentrations are ambient air temperature, relative humidity and vegetation density.

Specific comments:

1. Introduction: glucose is recognized as a tracer for plant pollen but also for biomass burning. I miss some discussion about this issue in the introduction.

2. In several parts of the text, figures and tables, mention is made of "glucose" but in fact "free cellulose" is meant. In order to avoid confusion, I suggest to make this more clear and replace "glucose" by "free cellulose".

Technical corrections:

References: should be ordered chronologically.

---

## Referee Comment (RC2) · Anonymous Referee #3 · 20 Jun 2019

The manuscript "Arabitol, mannitol and glucose as tracers of primary biogenic organic aerosol: influence of environmental factors on ambient air concentrations and spatial distribution over France" describes the primary sugar compounds (SC, defined as glucose, arabitol and mannitol) concentrations in PM10 for 16 increasing space scale sites (local to nation-wide), distributed in several French geographic areas of different environmental conditions. This paper first time investigates the spatial behavior of these chemicals and evidencing their major effective environmental drivers.

Major comments:

(1) Updating the references used in this manuscript to more current state is suggested.

(2) LOD (limit of detection) of the detected chemicals should be included in the experimental section.

(3) The regional transport is also very important for the spatial behavior and distribution of the chemical species in the ambient. In addition, only temporal variations and tracer ratios were shown and discussed in the results and discussion section. More deep analysis (i.e., the influences from nearby regions/sources, combine the chemical analysis results with synoptic data,. . .) are recommended to make this paper more interesting and innovative. At least, choose one or two cases to explain the contribution from regional transport by backward trajectory analysis.

(4) Page 6, Line 174-180. The normalized cross-correlation (NCC) test was chosen in this manuscript, and author mentioned a thorough discussion on the normalized cross-correlation method can be found elsewhere (Kaso, 2018; Yoo and Han, 2009). However, there was no related applied reference of NCC method was given, more field observation references used this methods are suggested to add.

(5) The lines in the figures are too thick to find the points, especially for Fig. (2a), Fig. (2b) and Fig. 5. It is difficult to separate the different color lines. Moreover, the thickness of the lines seems not consistent, i.e., the blue lines seem thicker than other color lines.

(6) Figure S2 is suggested to add in Figure 3. Discuss the Normalized cross-correlation values for the daily evolution of particulate for glucose, polyols, calcium and sulfate together. It can exhibit the differences of NCC between these chemicals more directly. Moreover, how about NCC of other inorganic ions, i.e., $NH_4^+$, $NO_3^-$ (similar as $SO_4^{2-}$, are the main components of secondary inorganic aerosols), $K^+$ (biomass burning tracer) and $Cl^-$.

Specific comments:

(1) Line 190: The linear regression (lm) package in R was employed for multiple regression analyses. What does "lm" in the bracket means??

(2) Line 320: these findings highlight that particulate SC PM10 and cellulose in both urban background and rural agricultural areas..., should be changed to "these findings highlight that SC in PM10 and cellulose in both urban background and rural agricultural areas"

---

## Referee Comment (RC3) · Anonymous Referee #2 · 26 Jun 2019

This paper describes the evolutions of glucose, mannitol and arabitol in the aerosol covering 16 sites all over France. The study consists in a huge and precious dataset. For the first time, the distance-dependent correlation is demonstrated, investigating also the main drivers of atmospheric sugar concentrations.

General comment: please check all manuscript, including figures and tables, and modify the term "polyols" with "mannitol and arabitol", as necessary, to avoid confusion., as suggested in the initial revision.

[Figure]

Line 41-43. The authors affirmed that "sugar alcohols . . .- including arabitol and mannitol. . .- have been recognized as tracers for airborne fungi". One of the main objectives of my recent research is the source investigation of water soluble organic compounds, such as for example sugars, and I quite sure that some sugars alcohols have another source. For examples I saw that sorbitol have some correlation with biomass burning tracers, while arabitol and mannitol, mainly distributed in the coarse fraction of aerosol, plausibly originate from fungal spores. So, I suggest to focus your affirmation only on the arabitol and mannitol.

Line 45. The authors define glucose "a specific tracer for plant materials" but I think that the authors should remove "specific" because glucose can have different sources: plant materials, soil emissions (as suggested by the authors) and also marine biogenic material derived from degradation of polysaccharides present in the marine microlayer. I suggest to read some papers of Prof. Leck because she investigated the organic compounds (such as saccharides) in the marine aerosol. I know that the paper is focused on the aerosol samples collected in the areas far from the coast but in the introduction I think that the authors should consider all sources.

Line 174. Can you specify some details about the dataset matrix using to perform the normalized cross-correlation.

Line 257. You correctly affirmed that mannitol-to arabitol ratio can suggest the tempora and spatial evolution of their amission processes, using this reference: Gosselin et al. 2016. This paper demonstrated also that, in some cases, mannitol and arabitol can have different sources: "mannitol is a commonpolyol in higher plants while arabitol is only found in fungal spores and lichen". I suggest to insert this concept in the manuscript and to consider the $R^2$ between two polyols in the discussion because maybe either conclusion can be also obtained (this is just a suggestion).

Section 3.2. The distance-dependent correlations and the SC evolution synchronous at an urban city scale and throughout the same geographical regions are the very

interesting topics in the manuscript and I appreciate this work because it was a lack of the sugars knowledge. The distance-dependent correlations is very clear using your approach but I suggest to clarify the main reasons for the decrease of NCC when the distance was above 200 km. You report some explanations but I suggest to deeply discuss the reasons or the suggestion of this behavior.

Line 301. Please remove "s" from "corresponds".

---

## Author Comment (AC1) · 16 Jul 2019

ACP-2019-434 Answer to Anonymous Referee #1 comments:

This manuscript provides detailed insights into the biogenic primary organic aerosol emission sources of the primary sugar compounds (SC), i.e., glucose, arabitol and mannitol. The study has been carefully designed and the results have been interpreted in detail. The study covers 16 nation-wide sites all over France and contains a very comprehensive data set. It is clearly shown that the main drivers of SC atmospheric

concentrations are ambient air temperature, relative humidity and vegetation density.

We thank the reviewer for his/her review. We have studied the comments and we have made revisions point by point. The detailed responses to the comments are given below, point by point, in blue color, including changes directly made to the manuscript, in red color.

Specific comments:

1. Introduction: glucose is recognized as a tracer for plant pollen but also for biomass burning. I miss some discussion about this issue in the introduction.

The reviewer is right that glucose can also originate from the thermal degradation of the plant materials (e.g., cellulose, a polymer of glucose). However, as evidenced in Figure R1.1, the concentrations of levoglucosan (a well-established tracer of biomass burning source) and those of glucose clearly display very different annual atmospheric evolution cycles: higher concentrations of levoglucosan in France are observed in the coldest season (winter) due to the increased biomass burning while those of glucose are observed in in warm seasons and coinciding with negligible ambient concentrations of levoglucosan. Such different temporal patterns indicate that the biomass burning is not an important source of atmospheric glucose.

Glucose can have a broad primary biogenic sources, e.g. from terrestrial plant pollen, fruits, and detritus, or from the degradation of the soil microorganisms (Xiao et al., 2018; Zhu et al., 2015) or even possibly from bubble bursting processes in remote oceans (Fu et al., 2013; Gao et al., 2011; Leck and Bigg, 2005). For these reasons, we have removed the term "specific" in lines 54-59.

2. In several parts of the text, figures and tables, mention is made of "glucose" but in fact "free cellulose" is meant. In order to avoid confusion, I suggest to make this more clear and replace "glucose" by "free cellulose".

In fact, both glucose and free cellulose are measured and analyzed separately in the

present work. We used glucose when the monosaccharide "glucose" is meant and free cellulose we considered the cellulose ambient cellulose.

Technical corrections: References: should be ordered chronologically. The references are now ordered chronologically, as suggested by the reviewer.

References Fu, P. Q., Kawamura, K., Chen, J., Charrière, B., and Sempéré, R.: Organic molecular composition of marine aerosols over the Arctic Ocean in summer: contributions of primary emission and secondary aerosol formation, Biogeosciences, 10(2), 653–667, doi:10.5194/bg-10-653-2013, 2013. Gao, Q., Nilsson, U., Ilag, L. L., and Leck, C.: Monosaccharide compositional analysis of marine polysaccharides by hydrophilic interaction liquid chromatography-tandem mass spectrometry, Anal. Bioanal. Chem., 399(7), 2517–2529, doi:10.1007/s00216-010-4638-z, 2011. Leck, C. and Bigg, E. K.: Biogenic particles in the surface microlayer and overlaying atmosphere in the central Arctic Ocean during summer, Tellus B, 57(4), 305–316, doi:10.1111/j.1600-0889.2005.00148.x, 2005. Xiao, M., Wang, Q., Qin, X., Yu, G., and Deng, C.: Composition, Sources, and Distribution of PM2.5 Saccharides in a Coastal Urban Site of China, Atmosphere, 9(7), 274, doi:10.3390/atmos9070274, 2018. Zhu, C., Kawamura, K., and Kunwar, B.: Organic tracers of primary biological aerosol particles at subtropical Okinawa Island in the western North Pacific Rim: Organic biomarkers in the north pacific, J. Geophys. Res. Atmospheres, 120(11), 5504–5523, 2015.

Please also note the supplement to this comment: https://www.atmos-chem-phys-discuss.net/acp-2019-434/acp-2019-434-AC1-supplement.pdf

[Figure]

[Figure]

**Fig. 1.** Annual evolution cycles of the glucose (left) and levoglucosan (right) concentrations in PM10 measured at the urban site of Grenoble Les Frênes, from the years 2012 to 2018 (details in supplement)

---

## Author Comment (AC2) · 16 Jul 2019

ACP-2019-434 Answer to Anonymous Referee #2 comments

The manuscript "Arabitol, mannitol and glucose as tracers of primary biogenic organic aerosol: influence of environmental factors on ambient air concentrations and spatial distribution over France" describes the primary sugar compounds (SC, defined as glucose, arabitol and mannitol) concentrations in PM10 for 16 increasing space scale sites (local to nation-wide), distributed in several French geographic areas of different environmental conditions. This paper first time investigates the spatial behavior of these chemicals and evidencing their major effective environmental drivers. We thank the reviewer for his/her attention to our manuscript that greatly contribute to improve the quality of this research paper. All comments have been considered and answered. The detailed responses to the comments are given below, point by point, in blue color, including changes directly made to the manuscript, in red color.

Major comments:

(1) Updating the references used in this manuscript to more current state is suggested.

We do agree with the reviewer and we have updated the references with several works recently published, including those in 2019. However, very few scientific papers have been published recently on the short term (daily) and the spatial characterization of polyols and glucose in PM10. This is why older pioneering works are also cited in the present work.

(2) LOD (limit of detection) of the detected chemicals should be included in the experimental section.

As suggested by reviewer, the information about the quantification limits have been included in the experimental section (lines 176-177).

(3) The regional transport is also very important for the spatial behavior and distribution of the chemical species in the ambient. In addition, only temporal variations and tracer ratios were shown and discussed in the results and discussion section. More deep analysis (i.e., the influences from nearby regions/sources, combine the chemical analysis results with synoptic data,. . .) are recommended to make this paper more interesting and innovative. At least, choose one or two cases to explain the contribution from regional transport by backward trajectory analysis.

We agree that regional transport may impact PM polyol concentrations. However, we do not think that it explains the main temporal signals observed in this work. Since the

correlation matrix corresponds to averaged values of composite data, i.e. aggregate on consecutive three days or six days intervals, it already account for potential regional transport between sites, and a decreased of correlation with inter-site distance is observed and is so probably indicative of local source contribution rather than transportation. However, as suggested by the reviewer, we achieved additional back trajectory analyses. , This was done for arabitol concentrations at the remote OPE-ANDRA site for the period 2012−2018, applying Potential Source Contribution Function (PSCF) to HYSPLIT data and using the pyPSCF python package . Results do not indicate clear source region(s) (Figure R2.1). Indeed, even if it seems that air-masses associated with high arabitol loading (>75th concentration percentile) never come from the East, it is in fact explained by the climatic wind condition in this region where no easterlies wind are observed during summer (anticyclonic condition). Finally, since no specific region is pointed out by the PSCF analysis, it may be explain either if the arabitol is emitted everywhere, or by a local (<few grid cells, within around 100 km from the station) source since all back-trajectories will then be associated with high concentration. Since correlations between sites decrease with the distance, the first hypothesis is most probably not valid. These two arguments are in favor of local sources being predominant for the polyols, as opposed to regional (> 100 km) transport.

(4) Page 6, Line 174-180. The normalized cross-correlation (NCC) test was chosen in this manuscript, and author mentioned a thorough discussion on the normalized cross-correlation method can be found elsewhere (Kaso, 2018; Yoo and Han, 2009). However, there was no related applied reference of NCC method was given, more field observation references used this methods are suggested to add.

The reviewer is right. The references related to the NCC method described only the concept and theory of NCC method. Sorry about it. Additional references (Bardal and Sætran, 2016; Dai and Zhou, 2017; Eisner et al., 2009; Lainer et al., 2016; Le Pichon et al., 2019) are now given to illustrate NCC applications in atmospheric sciences (lines 206-207).

(5) The lines in the figures are too thick to find the points, especially for Fig. (2a), Fig. (2b) and Fig. 5. It is difficult to separate the different color lines. Moreover, the thickness of the lines seems not consistent, i.e., the blue lines seem thicker than other color lines.

These figures have been modified accordingly. Thanks for suggestion.

(6) Figure S2 is suggested to add in Figure 3. Discuss the Normalized cross-correlation values for the daily evolution of particulate for glucose, polyols, calcium and sulfate together. It can exhibit the differences of NCC between these chemicals more directly. Moreover, how about NCC of other inorganic ions, i.e., NH4+, NO3- (similar as SO42- , are the main components of secondary inorganic aerosols), K+ (biomass burning tracer) and Cl-.

The present work do not aim at discussing these species. Nevertheless, since they may act as a negative control for the local emission of polyols, we initially presented some of them in the submitted manuscript and SI. As suggested by the reviewer, former Fig S2 has also added in the main text together with fig 3. and we are now presenting some major secondary inorganics (ammonium) and biomass burning proxy (levoglucosan) as follows:

Lines 266-279 "Unlike SC, ambient air concentrations of sulfate (Fig. 3C) and ammonium (Fig. 3D), associated with long-range aerosol transport (Abdalmogith and Harrison, 2005; Amato et al., 2016; Coulibaly et al., 2015; Pindado and Perez, 2011; Waked et al., 2014) and levoglucosan ((Fig. 3E), associated with biomass burning in cold season (Weber et al., 2019; Xiao et al., 2018), display stronger positive correlations (R > 0.72-0.98, p < 0.01) at all pairs of sites considered in the present work. Moreover, ambient concentrations of calcium (Fig. 3F), associated with local fugitive dust sources or/and long-range aerosol transport (Ram et al., 2010; Wan et al., 2019) display random correlation patterns. These results are in agreement with Zhu et al. (2018) who also reported non-significant correlations between SC and sulfate in PM2.5

aerosols measured at Shanghai, China. The distinct spatial behaviors between sulfate (or Ca2+) and SC in the present work further suggest a dominant regional influence for atmospheric SC, as opposed to processes associated with either local sources for calcium or long-range transport for sulfate".

For secondary species (sulfate and ammonium), potential long range transport (Ca2+) and chemically stable species (levoglucosan, Figure R2.1B), the correlation are still high (r>0.7) even after hundreds of kilometers. For these species, we can make the hypothesis that the regional transport play a major role in concentrations seen at a given site. However, an in-depth analysis of the sources and evolution of the concentrations of these species is beyond the scope of this study and would require a dedicated future work.

Specific comments:

(1) Line 190: The linear regression (lm) package in R was employed for multiple regression analyses. What does "lm" in the bracket means??

The linear model aka "lm" in the brackets is the name of the statistical package employed for multiple regression analyses. Definition of "lm" is now added in the main text (line 217).

(2) Line 320: these findings highlight that particulate SC PM10 and cellulose in both urban background and rural agricultural areas. . ., should be changed to "these findings highlight that SC in PM10 and cellulose in both urban background and rural agricultural areas

Thank you for your attentive review, this sentence has been corrected (lines 367-368).

Please also note the supplement to this comment:
https://www.atmos-chem-phys-discuss.net/acp-2019-434/acp-2019-434-AC2-supplement.pdf

[Figure]

OPE, Arabitol > 42.1316
From 2012/01/10 to 2018/10/29

A)

OPE, Levoglucosan > 151.64704
From 2012/01/10 to 2018/10/29

B)

OPE
Backtrajectories probability (log(n))

C)

**Fig. 1.** PSCF analysis for the OPE site (using pyPSCF and HYSPLIT). Back-trajectories associated with arabitol concentrations higher than the 75th percentile divided by the number of back-trajectories.

[Figure]

---

## Author Comment (AC3) · 16 Jul 2019

ACP-2019-434 Answer to Anonymous Referee #3 comments

This paper describes the evolutions of glucose, mannitol and arabitol in the aerosol covering 16 sites all over France. The study consists in a huge and precious dataset. For the first time, the distance-dependent correlation is demonstrated, investigating also the main drivers of atmospheric sugar concentrations.

We thank the reviewer for his/her attention to our manuscript that greatly contribute to

improve the quality of this research paper. We have considered each of the comments and we have made revisions point by point. The detailed responses to the comments are given below, point by point, in blue color, including changes directly made to the manuscript, in red color.

General comment: please check all manuscript, including figures and tables, and modify the term "polyols" with "mannitol and arabitol", as necessary, to avoid confusion., as suggested in the initial revision.

We do agree with the reviewer that we are not analyzing all the sugar alcohol species. However, we clearly specify that the term polyols is to refer to the sum of arabitol and mannitol concentrations (lines 178-179). This has been added to the main text: "Hereafter, the term "Polyols" is used to refer uniquely to the sum of arabitol and mannitol concentrations".

Line 41-43. The authors affirmed that "sugar alcohols . . .- including arabitol and mannitol. . .- have been recognized as tracers for airborne fungi". One of the main objectives of my recent research is the source investigation of water soluble organic compounds, such as for example sugars, and I quite sure that some sugars alcohols have another source. For examples I saw that sorbitol have some correlation with biomass burning tracers, while arabitol and mannitol, mainly distributed in the coarse fraction of aerosol, plausibly originate from fungal spores. So, I suggest to focus your affirmation only on the arabitol and mannitol.

We do agree with the reviewer and we have focused our affirmation only on the arabitol and mannitol. Line 45. The authors define glucose "a specific tracer for plant materials" but I think that the authors should remove "specific" because glucose can have different sources: plant materials, soil emissions (as suggested by the authors) and also marine biogenic material derived from degradation of polysaccharides present in the marine microlayer. I suggest to read some papers of Prof. Leck because she investigated the organic compounds (such as saccharides) in the marine aerosol. I know that the paper

is focused on the aerosol samples collected in the areas far from the coast but in the introduction I think that the authors should consider all sources.

Indeed, glucose can have a broad of biogenic sources, e.g. from terrestrial plant pollen, fruits, and detritus, or from the degradation of the soil microorganisms (Kang et al., 2018; Li et al., 2018; Xiao et al., 2018) or even possibly from bubble bursting processes in remote oceans (Fu et al., 2013; Gao et al., 2011; Leck and Bigg, 2005); we have removed the term "specific" (see lines 54—59). This point is also further discussed in the response to the comment 1 of anonymous referee # 1.

Line 174. Can you specify some details about the dataset matrix using to perform the normalized cross correlation.

The raw data used in the present study consisted in to daily (24 hours) aerosols collected at 16 sites in different geographic regions in France. For pairwise normalized cross-correlation analyses, original daily series were first converted as follows: starting on identical days (for each pairs of sites), arrangement on the original daily data into consecutive 3-day intervals (or 6-day intervals in the case of OPE-ANDRA) and calculation of the average concentration values for the middle-day were performed. We directly used this resultant data for the correlation analysis between site pairs (lines 207-212).

In this respect, the manuscript has been revised as follows: lines 207-212 "To achieve pair-wise correlation analysis between the sampling sites collected during the same periods, the original raw daily measurements were processed as follows: starting on identical days for each pairs of sites, arrangement on the original daily data into consecutive 3-day intervals (or 6-day intervals in the case of OPE-ANDRA) and calculation of the average concentration values for the middle-day were performed. The resultant data were used for correlation analysis between site pairs (Table S3)".

Line 257. You correctly affirmed that mannitol-to arabitol ratio can suggest the temporal and spatial evolution of their emission processes, using this reference: Gosselin

et al. 2016. This paper demonstrated also that, in some cases, mannitol and arabitol can have different sources: "mannitol is a common polyol in higher plants while arabitol is only found in fungal spores and lichen". I suggest to insert this concept in the manuscript and to consider the R2 between two polyols in the discussion because maybe either conclusion can be also obtained (this is just a suggestion).

Thank you for this interesting suggestion. We added discussion about R2 between arabitol and mannitol (lines 288—292), as follows: lines 282-293.

"Based on parallel measurements of spore counts and PM10 polyol concentrations at three sites within the area of Vienna (Austria), Bauer et al. (2008a) found an average arabitol and mannitol content per fungal spores of respectively 1.2 pg spore-1 (range 0.8-1.8 pg spore-1) and 1.7 pg spore-1 (range 1.2-2.4 pg spore-1). Mannitol and arabitol have also been often identified in the green algae and lower plants (Buiarelli et al., 2013; Di Filippo et al., 2013; Vélëz et al., 2007; Xu et al., 2018; Zhang et al., 2010). Gosselin et al., 2016 observed a relatively low (R2 = 0.31) to high (R2 = 0.84) coefficient of determination between mannitol and arabitol for total suspended particles (TSP) collected at a pine-forested area during dry and rainy periods, respectively. High correlation in rainy periods possibly suggested that both chemical species in the TSP fraction in this pine-forested area could have been derived mainly from the same sources, i.e., actively wet-discharged ascospores and basidiospores, while the poor correlation in dry periods could have been likely due to more complex sources, i.e., dry discharged spores, plants, algae, etc."

Note that the study by Gosselin et al., 2016 has been conducted on total suspended particles (TSP) and in a specific pine-forested area of North America. The high coefficient of determination reported by Gosselin et al., 2016 during the rainy periods possibly suggest that both chemical species in the TSP fraction in this pine-forested area could have been derived mainly from the same sources, i.e., actively wet-discharged ascospores and basidiospores. However, the relatively poor correlation in dry periods could have been likely due to more complex sources, i.e., a mixture of actively

wet and/or dry discharged spores, or influence of additional biogenic sources such as plants, algae, etc. Indeed, active release of wet discharged ascospores and basidiospores occurs in most ascomycetes and in basidiomycetes (Ingold and Hudson, 1993; Zhang et al., 2010) which is influenced by ambient humidity and rainfall (Elbert et al., 2007; Zhang et al., 2015). In contrast, dry discharged spores are preferentially emitted under dry and warm conditions. Thus, these correlation patterns could be at least partially explained by the different fungal habitats and/or different emission processes during rainy and dry periods.

Section 3.2. The distance-dependent correlations and the SC evolution synchronous at an urban city scale and throughout the same geographical regions are the very interesting topics in the manuscript and I appreciate this work because it was a lack of the sugars knowledge. The distance-dependent correlations is very clear using your approach but I suggest to clarify the main reasons for the decrease of NCC when the distance was above 200 km. You report some explanations but I suggest to deeply discuss the reasons or the suggestion of this behavior.

Thank for this positive comment. We also believe that this point is quite innovative in the current literature and that it can considerably improve our knowledge about primary sugars in the atmosphere. We believe that the main reasons of such distance correlation patterns are most probably associated with different airborne microbial community assemblies that are shaped by different regional environmental factors (e.g. meteorological conditions, vegetation types and cover, etc.). Indeed, our recent interdisciplinary work (submitted for publication in "Science advances") has shown that the atmospheric concentration dynamics of polyols and some major saccharides (trehalose, glucose) are driven by only a few specific airborne fungal and bacterial genera. Further analyses for sites located in different climatic regions of France have also shown that airborne microbial assemblies associated with these chemical species vary regionally (unpublished data). This makes sense since different biotopes (meteorological conditions, vegetation types and cover) harbor distinct microbial communities (Bowers et al.,

2012; Liu et al., 2019).

Line 301. Please remove "s" from "corresponds". Thank you for your careful review, we have removed this "s".

References: Bowers, R. M., McCubbin, I. B., Hallar, A. G. and Fierer, N.: Seasonal variability in airborne bacterial communities at a high-elevation site, Atmos. Environ., 50, 41–49, doi:10.1016/j.atmosenv.2012.01.005, 2012. Elbert, W., Taylor, P. E., Andreae, M. O. and Pöschl, U.: Contribution of fungi to primary biogenic aerosols in the atmosphere: wet and dry discharged spores, carbohydrates, and inorganic ions, Atmospheric Chem. Phys., 7(17), 4569–4588, doi:10.5194/acp-7-4569-2007, 2007. Fu, P. Q., Kawamura, K., Chen, J., Charrière, B. and Sempéré, R.: Organic molecular composition of marine aerosols over the Arctic Ocean in summer: contributions of primary emission and secondary aerosol formation, Biogeosciences, 10(2), 653–667, doi:10.5194/bg-10-653-2013, 2013. Gao, Q., Nilsson, U., Ilag, L. L. and Leck, C.: Monosaccharide compositional analysis of marine polysaccharides by hydrophilic interaction liquid chromatography-tandem mass spectrometry, Anal. Bioanal. Chem., 399(7), 2517–2529, doi:10.1007/s00216-010-4638-z, 2011. Gosselin, M. I., Rathnayake, C. M., Crawford, I., Pöhlker, C., Fröhlich-Nowoisky, J., Schmer, B., Després, V. R., Engling, G., Gallagher, M., Stone, E., Pöschl, U. and Huffman, J. A.: Fluorescent bioaerosol particle, molecular tracer, and fungal spore concentrations during dry and rainy periods in a semi-arid forest, Atmospheric Chem. Phys., 16(23), 15165–15184, doi:10.5194/acp-16-15165-2016, 2016. Ingold, T. C. and Hudson, H. J.: The biology of fungi, Springer Netherlands. [online] Available from: doi10.1007/978-94-011-1496-7, 1993. Kang, M., Fu, P., Kawamura, K., Yang, F., Zhang, H., Zang, Z., Ren, H., Ren, L., Zhao, Y., Sun, Y. and Wang, Z.: Characterization of biogenic primary and secondary organic aerosols in the marine atmosphere over the East China Sea, Atmospheric Chem. Phys. Discuss., 1–45, doi:10.5194/acp-2018-318, 2018. Leck, C. and Bigg, E. K.: Biogenic particles in the surface microlayer and overlaying atmosphere in the central Arctic Ocean during summer, Tellus B, 57(4), 305–316, doi:10.1111/j.1600-

0889.2005.00148.x, 2005. Li, Y.-C., Shu, M., Ho, S. S. H., Yu, J.-Z., Yuan, Z.-B., Wang, X.-X., Zhao, X.-Q. and Liu, Z.-F.: Effects of Chemical Composition of PM2.5 on Visibility in a Semi-Rural City of Sichuan Basin, Aerosol Air Qual. Res., 18(4), 957–968, 2018. Liu, H., Hu, Z., Zhou, M., Hu, J., Yao, X., Zhang, H., Li, Z., Lou, L., Xi, C., Qian, H., Li, C., Xu, X., Zheng, P. and Hu, B.: The distribution variance of airborne microorganisms in urban and rural environments, Environ. Pollut., 247, 898–906, doi:10.1016/j.envpol.2019.01.090, 2019. Xiao, M., Wang, Q., Qin, X., Yu, G. and Deng, C.: Composition, Sources, and Distribution of PM2.5 Saccharides in a Coastal Urban Site of China, Atmosphere, 9(7), 274, doi:10.3390/atmos9070274, 2018. Zhang, T., Engling, G., Chan, C.-Y., Zhang, Y.-N., Zhang, Z.-S., Lin, M., Sang, X.-F., Li, Y. D. and Li, Y.-S.: Contribution of fungal spores to particulate matter in a tropical rainforest, Environ. Res. Lett., 5(2), 024010, doi:10.1088/1748-9326/5/2/024010, 2010. Zhang, Z., Engling, G., Zhang, L., Kawamura, K., Yang, Y., Tao, J., Zhang, R., Chan, C. and Li, Y.: Significant influence of fungi on coarse carbonaceous and potassium aerosols in a tropical rainforest, Environ. Res. Lett., 10(3), 034015, doi:10.1088/1748-9326/10/3/034015, 2015.

Please also note the supplement to this comment:
https://www.atmos-chem-phys-discuss.net/acp-2019-434/acp-2019-434-AC3-supplement.pdf
* * *

---

## Author Response (AR2)

Institut des Géosciences de l'Environnement

Dr Jean-Luc Jaffrezo  (Jean-Luc.Jaffrezo@univ-grenoble-alpes.fr )

Abdoulaye Samaké (abdoulaye.samake2@univ-grenoble-alpes.fr )

460, Rue de la Piscine

BP 53, Domaine Universitaire

38041 Grenoble, France

[Figure]

Grenoble, August 3, 2019.

Dear Pr Alex Huffman

Thank you for your consideration, please find enclosed our revised manuscript on "Arabitol, mannitol and glucose as tracers of primary biogenic organic aerosol: influence of environmental factors on ambient air concentrations and spatial distribution over France" by Abdoulaye Samaké et al. (MS No.: ACP-2019-434).

Thank you for your constructive technical comments that contribute to improve this work.

Concerning your comment about our recent work submitted for publication in Sci. Advances, mentioned in response to "the last big comment from referee #3", you are right that this work was not referenced. Sorry about it. Please, find below a reference of this publication (still under review).

Samaké A., A. Bonin, J.-L. Jaffrezo, P. Taberlet, G. Uzu, S. Conil, and J.M.F. Martins. (2019): High levels of primary biogenic organic aerosols in the atmosphere in summer are driven by only a few microorganisms from the leaves of surrounding plants, Sci. Advances, In Review.

Additionally, as illustrate in the marked-up manuscript version (see below), we have considered all of your technical comments and we made corrections accordingly.

I therefore resubmit our paper after considering all remarks and I truly hope that our improvements will meet your requirements.

On behalf all co-authors,

[revised manuscript text omitted]